# Beyond Efficiency: Molecular Data Pruning for Enhanced Generalization

**Dingshuo Chen**[1,2]    **Zhixun Li**[3]    **Yuyan Ni**[4]    **Guibin Zhang**[5]    **Ding Wang**[1,2]
**Qiang Liu**[1,2]    **Shu Wu**[1,2*]    **Jeffrey Xu Yu**[3]    **Liang Wang**[1,2]

[1]New Laboratory of Pattern Recognition
State Key Laboratory of Multimodal Artificial Intelligence Systems
Institute of Automation, Chinese Academy of Sciences
[2] School of Artificial Intelligence, University of Chinese Academy of Sciences
[3]Department of Systems Engineering and Engineering Management
The Chinese University of Hong Kong
[4]Academy of Mathematics and Systems Science, Chinese Academy of Sciences
[5]Tongji University
✉ Primary contact: `dingshuo.chen@cripac.ia.ac.cn`

## Abstract

With the emergence of various molecular tasks and massive datasets, how to perform efficient training has become an urgent yet under-explored issue in the area. Data pruning (DP), as an oft-stated approach to saving training burdens, filters out less influential samples to form a coreset for training. However, the increasing reliance on pretrained models for molecular tasks renders traditional in-domain DP methods incompatible. Therefore, we propose a Molecular data Pruning framework for enhanced Generalization (`MolPeg`), which focuses on the *source-free data pruning* scenario, where data pruning is applied with pretrained models. By maintaining two models with different updating paces during training, we introduce a novel scoring function to measure the informativeness of samples based on the loss discrepancy. As a plug-and-play framework, `MolPeg` realizes the perception of both source and target domain and consistently outperforms existing DP methods across four downstream tasks. Remarkably, it can surpass the performance obtained from full-dataset training, even when pruning up to 60-70% of the data on HIV and PCBA dataset. Our work suggests that the discovery of effective data-pruning metrics could provide a viable path to both **enhanced efficiency** and **superior generalization** in transfer learning.

## 1 Introduction

The research enthusiasm for developing molecular foundation models is steadily increasing [1–5], attributed to its foreseeable performance gains with ever-larger model and amounts of data, as observed neural scaling laws [6, 7] and emergence ability [8] in other domains. However, the computational and storage burdens are daunting in model training [9], hyperparameter tuning, and model architecture search [10–12]. It is therefore urgent to ask for training-efficient molecular learning in the community.

Data pruning (DP), in a natural and simple manner, involves the selection of the most influential samples from the entire training dataset to form a coreset as *paragons* for model training. The primary goal is to alleviate training costs by striking a balance point between efficiency and performance compromise. A trend in this field is developing data influence functions [13–15], training dynamic metrics [16–19], and coreset selection [20–22] for lossless - although typically compromised - model

---

*Corresponding author: Shu Wu (`shu.wu@nlpr.ia.ac.cn`).

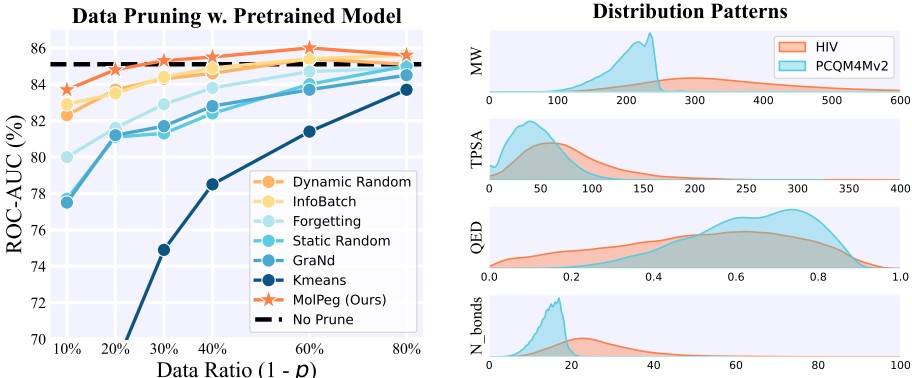

Figure 1: **(Left)** The performance comparison of different data pruning methods in HIV dataset under source-free data pruning setting. **(Right)** Distribution patterns of four important molecular features - molecular weight (MW), topological polar surface area (TPSA), Quantitative Estimate of Drug-likeness (QED) and number of bonds - in PCQM4Mv2 [33] and HIV [34] dataset, which are used for pretraining and finetuning, respectively.

generalization. When it comes to molecular tasks, transfer learning, particularly the ***pretrain-finetune*** paradigm, has been regarded as the de-facto standard for enhanced training stability and superior performance [23–27]. However, existing DP methods are purposed for train-from-scratch setting, i.e., the model is randomly initialized and trained on the selected coreset. A natural question arises as to *whether or not current DP methods remain effective when applied with pre-trained models*. Experimental analysis, as illustrated in Figure 1 (Left), suggests a negative answer. Most existing pruning strategies exhibit inferior results relative to the performance achieved with the full dataset, even falling short of simple random pruning.

In contrast to the existing DP approaches, which focus solely on a single target domain, the incorporation of pretrained model introduces an additional source domain, thereby inevitably exposing us to the challenge of distribution shift [28, 29]. Unfortunately, this is especially severe in molecular tasks, owing to the limited diversity of large-scale pretraining datasets compared to the varied nature of downstream tasks. As illustrated in Figure 1 (Right), we investigate the distribution patterns of several important molecular properties across the upstream and downstream datasets following Beaini et al. [30]. The observed disparities impede the model generalization, thus making DP with pretrained models a highly non-trivial task. We define this out-of-domain DP setting as ***source-free data pruning***. It entails removing data from downstream tasks leveraging pre-trained models while remaining agnostic to the specifics of the pre-training data.

Of particular relevance to this work are approaches that propose DP methods for transfer learning [31, 32], which also target cross-domain scenarios. Despite the promising results they achieved, these methods select pretraining samples based on downstream data distribution, which necessitates reevaluation of previously selected samples and retraining heavy models as new samples involving, undermining the goal of achieving generalization and universality in pretraining. To this end, we take a step towards designing a DP method under the source-free data pruning setting to achieve **efficient and effective** model training, which aligns better with practical deployment for molecular tasks.

In this work, we propose a Molecular data Pruning framework for enhanced Generalization, which we term `MolPeg` for brevity. The core idea of `MolPeg` is to achieve cross-domain perception via maintaining an online model and a reference model during training, which places emphasis on the target and source domain, respectively. Besides, we design a novel scoring function to simultaneously select easy (representative) and hard (challenging) samples by comparing the absolute discrepancy between model losses. We further take a deep dive into the theoretical understanding and glean insight on its connection with the previous DP strategies. Note that our proposed `MolPeg` framework is generic, allowing for seamless integration of off-the-shelf pretrained models and architectures. To the best of our knowledge, this is the first work that studies how to perform data pruning for molecular learning from a transfer learning perspective. Our contributions can be summarized as follows:

- We analyze the challenges of efficient training in the molecular domain and formulate a tailored DP problem for transfer learning, which better aligns with the practical requirements of molecular pre-trained models.

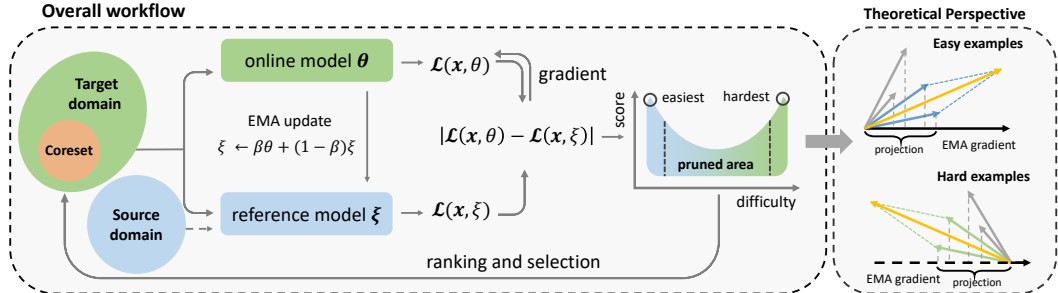

Figure 2: The overall framework of `MolPeg`. **(Left)** We maintain an online model and a reference model with different updating paces, which focus on the target and source domain, respectively. After model inference, the samples are scored by the absolute loss discrepancy and selected in ascending order. The easiest and hardest samples are given the largest score and selected to form the coreset. **(Right)** The selection process of `MolPeg` can be interpreted from a gradient projection perspective. Samples with low projection norms (grey) are discarded, while those with high norms are kept.

- We propose an efficient data pruning framework that can perceive both the source and target domains. It can achieve lightweight and effective DP without the need for retraining, facilitating easy adaptation to varied downstream tasks. We also provide a theoretical understanding of `MolPeg` and build its connections with existing DP strategies.

- We conduct extensive experiments on 4 downstream tasks, spanning different modalities, pretraining strategies, and task settings. Our method can surpass the full-dataset performance when up to 60%-70% of the data is pruned, which validates the effectiveness of our approach and unlocks a door to enhancing model generalization with fewer samples.

## 2 Preliminaries

In this section, we take a detour to revisit the traditional data pruning setting and *pretrain-finetune* paradigm before introducing the problem formulation of source-free data pruning.

**Problem statement of traditional data pruning**. Consider a learning scenario where we have a large training set denoted as $\mathcal{D} = \{(\boldsymbol{x_i}, y_i)\}_{i=1}^{|\mathcal{D}|}$, consisting of input-output pairs $(\boldsymbol{x_i}, y_i)$, where $\boldsymbol{x_i} \in \mathcal{X}$ represents the input and $y_i \in \mathcal{Y}$ denotes the ground-truth label corresponding to $\boldsymbol{x_i}$. Here, $\mathcal{X}$ and $\mathcal{Y}$ refer to the input and output spaces, respectively. The objective of traditional data pruning is to identify a subset $\hat{\mathcal{D}} \subset \mathcal{D}$, that captures the most informative instances. The model trained on this subset $\hat{\mathcal{D}}$ should yield a slightly inferior or comparative performance to the model trained on the entire training set $\mathcal{D}$. Thus they need to strike a balance between efficiency and performance.

**Revisit on transfer learning**. Given source and target domain datasets $\mathcal{D}_\mathcal{S}$ and $\mathcal{D}_\mathcal{T}$, the goal of pretraining is to obtain a high-quality feature extractor $f$ in a supervised or unsupervised manner. While in the finetuning phase, we aim to adapt the pretrained $f$ in conjunction with output head $g$ to the target dataset $\mathcal{D}_\mathcal{T}$.

Considering the proficiency of molecular pre-trained models in capturing meaningful chemical spaces, their widespread usage in enhancing performance across diverse molecular tasks has become commonplace. This necessitates a reassessment of the conventional approach to DP within the molecular domain and, more broadly, within the field of transfer learning. Previous attempts [31, 32] in data pruning for transfer learning primarily focus on trimming upstream data, selecting samples that closely match the distribution of downstream tasks to align domain knowledge. However, this necessitates retraining the model from scratch, which is notably ill-suited for the molecular domain, where the continual influx of new molecules introduces novel functionalities and structures. To this end, we propose a tailored DP problem for molecular transfer learning:

**Problem formulation** (Source-free data pruning). *Given a target domain dataset $\mathcal{D}_\mathcal{T}$ and a pretrained feature extractor parameterized by $\theta_\mathcal{S}$, we aim to identify a subset $\hat{\mathcal{D}}_\mathcal{T} \subset \mathcal{D}_\mathcal{T}$ for training, while being agnostic of the source domain dataset $\mathcal{D}_\mathcal{S}$, to maximize the model generalization.*

## 3 Methodology

As with generic data pruning pipelines, the `MolPeg` framework is divided into two stages, scoring and selection. In the first stage, we define a scoring function to measure the informativeness of samples and apply it to the training set. In the subsequent stage, given the sample scores, we rank them in ascending order and maintain the high-ranking samples for training. Note that our pruning method is dynamically performed during the training process, rather than conducted before training.

We next introduce the `MolPeg` framework in detail. We track the training dynamics of two models with different update paces. For each training sample, we measure the difference in loss between the two models to quantify its importance, and then make the final selection based on this metric. In the following parts, we first intuitively introduce our design of the scoring function. Then, we further explore the theoretical support behind the effectiveness of the `MolPeg`. The overall framework is illustrated in Figure 2 .

### 3.1 The `MolPeg` framework

The design of the scoring function addresses two key issues, (1) how to achieve the perception of source and target domain and (2) how to measure the informativeness of the samples.

**Cross-domain perception.** Since we are unable to access the upstream dataset, the pre-trained model serves as the only entry point of the source domain. During the finetuning stage, apart from *online encoder* undergoing gradient optimization via back-propagation, we further maintain a *reference encoder* updated with exponential moving average (EMA) to perceive the cross-domain knowledge. Note that both encoders are initialized by pretrained model $\theta_0 = \xi_0 = \theta^S$, where $\theta_t$ and $\xi_t$ denotes the parameters of online and reference model at batch step $t$, respectively. They are updated as follows:

$$\theta_{t+1} = \theta_t - \alpha \nabla_\theta \mathcal{L}(\hat{\mathcal{D}}_t, \theta_t) \quad \xi_t = \beta \theta_t + (1 - \beta)\xi_{t-1} \tag{1}$$

where $\alpha$ is the learning rate and $\beta \in [0, 1)$ is the pace coefficient that controls the degree of history preservation. Here $\hat{\mathcal{D}}_t$ is the selected finetuning dataset for epoch $t$, and $\nabla_\theta \mathcal{L}(\hat{\mathcal{D}}_t, \theta_t)$ denotes the average gradient $\frac{1}{|\hat{\mathcal{D}}_t|} \sum_{x_i \in \hat{\mathcal{D}}_t} \nabla_\theta \mathcal{L}(x_i, \theta_t)$ for short. Intuitively, We control the influence of target domain on the reference encoder via EMA. With a small update pace $\beta$, the online encoder prioritizes target domain, while the reference encoder emphasizes source domain.

**Informativeness measurement and selection.** By far we explicitly represent the inaccessible source domain knowledge with the help of the reference model, facilitating us to further quantify the informativeness of each sample in the cross-domain context. Our motivation for measuring the sample informativeness comes from a recent work that improves the neural scaling laws [35]. They suggest that the best pruning strategy depends on the amount of initial data. When the data volume is large, retaining the hardest samples yields better pruning results than retaining the easiest ones; the conclusion is the opposite when the data volume is small. This contrasts with the conclusion that only the hardest samples should be selected [16]. From an intuitive perspective, simple samples are more representative, allowing the model to adapt to downstream tasks more quickly, while hard samples are crucial for model generalization since they are

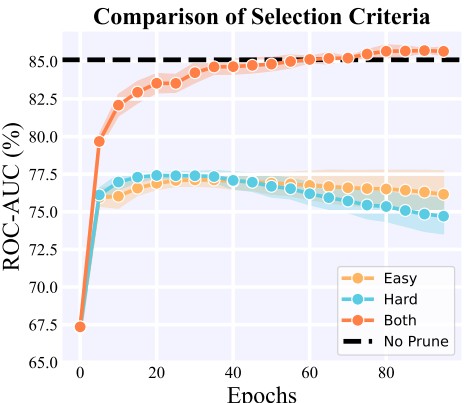

Figure 3: Performance comparison of selection criteria on HIV dataset when pruning 40% samples.

considered *supporting vectors* near the decision boundaries. This debate highlights that in data pruning, how to perform a mixture of easy and hard samples is a critical factor. As shown in Figure 3, when 60% samples in the HIV dataset are pruned, simply selecting the easiest or hardest samples leads to a performance drop in later epochs.

Therefore, we opt to retain both easy and hard samples instead of singularly removing one type. To measure the information gap between domains, we adopt both *online* and *reference encoder* to infer each sample and calculate the absolute loss discrepancy between them:

$$\hat{\mathcal{D}}_t = \{\boldsymbol{x} \in \mathcal{D}_t \mid |\mathcal{L}(\boldsymbol{x}, \theta_t) - \mathcal{L}(\boldsymbol{x}, \xi_t)| \geq \delta\}, \tag{2}$$

**Algorithm 1:** Molecular Data Pruning for Enhanced Generalization (`MolPeg`)

---

1 **Inputs:**

$\mathcal{D} = \{(\boldsymbol{x_i}, y_i, s_i)\}_{i=1}^{|\mathcal{D}|}$: dataset with the score for each example ($s_i = 1, \forall s_i \in \mathcal{D}$);
$\alpha$: learning rate; $\beta$: EMA update pace;
$p$: data pruning ratio ($p < 1$); $T$: total number of training epochs;
$f_\theta$: pretrained encoder parameterized by $\theta$

---

2 $t \leftarrow 0$;
3 **while** $t \leq T$ **do**
4     $K \leftarrow p \cdot |\mathcal{D}|$ ;               /* Get the number of remaining samples */
5     $\hat{\mathcal{D}}_t \leftarrow \text{TopK}(s)$ ;       /* Rank and Select the top-K samples for training */
6     $s_i \leftarrow \|\mathcal{L}(f_\theta(\boldsymbol{x_i}), y_i) - \mathcal{L}(f_\xi(\boldsymbol{x_i}), y_i)\|, \forall (\boldsymbol{x_i}, y_i, s_i) \in \hat{\mathcal{D}}_t$ ;     /* Scoring the samples */
7     $\theta \leftarrow \theta - \alpha \nabla_\theta \mathcal{L}(\hat{\mathcal{D}}_t, \theta)$ ;             /* Gradient update for online model */
8     $\xi \leftarrow \beta\theta + (1 - \beta)\xi$ ;              /* EMA update for reference model */
9     $t \leftarrow t + 1$
10 **return**

---

where $\mathcal{D}_t \in \mathcal{D}^{\mathcal{T}}$ comprises the target domain data sampled for batch step $t$ and $\hat{\mathcal{D}}_t \in \mathcal{D}_t$ comprises the data selected by `MolPeg`. $\delta$ is not a constant, but a threshold determined by the pruning ratio. Specifically, the rank of $\delta$ in the absolute loss discrepancy sequence $\{|\mathcal{L}(\boldsymbol{x}_i, \theta_t) - \mathcal{L}(\boldsymbol{x}_i, \xi_t)|\}_{i=1}^{|\mathcal{D}_t|}$ is $|\hat{\mathcal{D}}_t|$, i.e. pruning ratio×$|\mathcal{D}_t|$. It is easy to infer that a positive loss discrepancy, i.e. $\mathcal{L}(\boldsymbol{x}, \theta_t) - \mathcal{L}(\boldsymbol{x}, \xi_t) > 0$, indicates the model struggles to accurately distinguish the sample, identifying it as hard one. Conversely, a negative loss discrepancy indicates that the model can easily improve its accuracy, marking it as an easy sample. Therefore, intuitively, we dynamically assess the learning difficulty of samples during the training process. By measuring the absolute value of the loss discrepancy, we keep the simplest (most representative) and the hardest (most challenging) samples, which are integrated as the most informative ones (Orange line in Figure 3). We also provide the pseudo-code of `MolPeg` in Algorithm 1.

### 3.2 Theoretical Understanding

In this section, we explore the theoretical underpinnings of the data selection process in `MolPeg`. Recall that our scoring function is defined by loss discrepancy, we further make use of Taylor expansion on the designed scoring function. Then, from the gradient perspective, i.e., the first-order expansion term, we derived the following propositions and the complete proof is provided in the Appendix E.

**Assumption 1** (Slow parameter updating). *Assume the learning rate is small enough, so that the parameter update $\Delta\theta_t = \theta_{t+1} - \theta_t$ is small for every time step, i.e. $\|\Delta\theta_t\| \leq \epsilon$, $\forall t \in \mathbb{N}$, $\epsilon$ is a small constant.*

**Proposition 1** (Interpretation of loss discrepancy). *With Assumption 1, the loss discrepancy can be approximately expressed by the dot product between the data gradient and the "EMA gradient":*

$$\mathcal{L}(\boldsymbol{x}, \xi_t) - \mathcal{L}(\boldsymbol{x}, \theta_t) = \alpha \nabla_\theta \mathcal{L}(\boldsymbol{x}, \theta_t) \boldsymbol{v}_t^{EMA} + O(\epsilon^2), \tag{3}$$

*where $\boldsymbol{v}_t^{EMA}$ denotes $\sum_{j=1}^t (1 - \beta)^j \nabla_\theta \mathcal{L}(\hat{\mathcal{D}}_{t-j}, \theta_{t-j})$, i.e. the weighted sum of the historical gradients, which we termed as "EMA gradient".*

It indicates that the scoring function is essentially influenced by the magnitude of the dot product between the data gradient and the EMA gradient, as illustrated in Figure 2 (right). Given the EMA gradient, the size of the dot product is influenced by two factors: the norm of $\nabla_\theta \mathcal{L}(\boldsymbol{x}, \theta_t)$ and the angle between the two vectors. *(i)* A larger norm of the current data gradient is more likely to be selected, which resembles the criteria of GraNd score [19]. More connections to several well-known scoring functions are provided in the appendix F. (ii) If the current gradient direction closely aligns with the (opposite) EMA gradient direction, it often indicates an easy (hard) optimization of the sample, corresponding to the goal of selecting simple and hard samples in the previous analysis. Conversely, samples with gradient directions orthogonal to the EMA gradient are discarded.

In the following proposition, we examine the gradient of the selected samples and analyze simple and hard samples separately. Since the selection is performed at each fixed batch time step, we focus on one step of selection and omit the common time subscript $t$. Note that this result involves certain simplifications and approximations, and a formal version is provided in the appendix.

**Proposition 2** (Gradient projection interpretation of `MolPeg`, informal). *Let $\mathcal{D}^+ \subseteq \mathcal{D}$ and $\hat{\mathcal{D}}^+ \subseteq \hat{\mathcal{D}}$ denote the sets of samples for which the dot products between the data gradients and the "EMA gradient" are positive. Then, the gradient of the selected "simple" samples can be expressed as:*

$$\nabla_\theta \mathcal{L}(\hat{\mathcal{D}}^+, \theta) = \nabla_\theta \mathcal{L}(\mathcal{D}^+, \theta) + a v^{EMA}, a \geq 0. \tag{4}$$

*Similarly, we define $\mathcal{D}^- \in \mathcal{D}$ and $\hat{\mathcal{D}}^- \in \hat{\mathcal{D}}$ as samples that have negative dot products, then*

$$\nabla_\theta \mathcal{L}(\hat{\mathcal{D}}^-, \theta) = \nabla_\theta \mathcal{L}(\mathcal{D}^-, \theta) + b v^{EMA}, b \leq 0. \tag{5}$$

*$a = 0$ and $b = 0$ holds if and only if the loss discrepancy across $\mathcal{D}^+$ and $\mathcal{D}^-$ is uniform respectively, which are uncommon scenarios.*

Therefore, our data selection strategy essentially increases the weight of the (opposite) EMA gradient direction in the data gradient for easy (hard) samples. When $\mathcal{D}^+$ predominates, indicating a majority of simple samples in the dataset, this simplified model is akin to the momentum optimization strategy, which utilizes the sum of the current data gradient and the weighted EMA gradient to update the model parameters. This suggests that retaining simple samples may enhance optimization stability, allowing the model to overcome saddle points and local minima [36]. However, our method differs from the momentum optimization strategy in two key aspects. Firstly, we preserve directions opposite to the EMA gradient to target hard and forgettable samples. Secondly, our EMA gradient, which records the gradient of the coreset rather than the entire set, can retain more historical information under the same update pace.

# 4 Experimental Settings

## 4.1 Datasets and tasks

To comprehensively validate the effectiveness of our proposed `MolPeg`, we conduct experiments on three datasets, i.e., `HIV` [34], `PCBA` [37], `MUV` [38] and `QM9` [39], covering four types of molecular tasks. These tasks span two molecular modalities—2D graph and 3D geometry—as well as two types of supervised tasks, i.e., classification and regression.

Given the potential issues of over-fitting and spurious correlations that may arise with limited samples when a large pruning ratio is adopted, we focus on relatively large-scale datasets containing at least 40K molecules. Below, we briefly summarize the information of the datasets. For a more detailed description and statistics of the dataset, please refer to Appendix A.

## 4.2 Implementation details

In this section, we provide a succinct overview of the implementation details for our experiments, including backbone models for different modalities, training details and evaluation protocols.

**Backbone models.** Given the two modalities involved in our experiment, we need corresponding backbone models for data modeling. Below is a concise introduction to the backbone models. For a more comprehensive understanding of the model architecture, please refer to the Appendix D.

- For 2D graphs, we utilize the Graph Isomorphism Network (GIN) [40] as the encoder. To ensure the generalizability of our research findings, we adopt the commonly recognized experimental settings proposed by Hu et al. [41], with 300 hidden units in each layer, and a 50% dropout ratio. The number of layers is set to 5.

- For 3D geometries, we employ two widely used backbone models, PaiNN [42] and SchNet [43], as the encoders for different datasets. For SchNet, we set the hidden dimension and the number of filters in continuous-filter convolution to 128. The interatomic distances are measured with 50 radial basis functions, and we stack 6 interaction layers. For PaiNN, we adopt the setting with 128 hidden dimensions, 384 filters, 20 radial basis functions, and stack 3 interaction layers.

Table 1: The performance comparison to state-of-the-art methods on HIV and PCBA in terms of ROC-AUC (%, ↑) and Average Precision (%, ↑). We highlight the best-performing results in **boldface**. The performance difference with whole dataset training is highlighted with blue and orange, respectively.

| | Dataset | HIV | | | | | | PCBA | | | | | |
|---|---|---|---|---|---|---|---|---|---|---|---|---|---|
| | Pruning Ratio % | 90 | 80 | 70 | 60 | 40 | 20 | 90 | 80 | 70 | 60 | 40 | 20 |
| Static | Hard Random | $77.7_{\downarrow7.4}$ | $81.1_{\downarrow4.0}$ | $81.3_{\downarrow3.8}$ | $82.4_{\downarrow2.7}$ | $84.0_{\downarrow1.1}$ | $85.0_{\downarrow0.1}$ | $14.6_{\downarrow11.7}$ | $18.7_{\downarrow7.6}$ | $21.1_{\downarrow5.2}$ | $23.2_{\downarrow3.1}$ | $25.3_{\downarrow1.0}$ | $26.2_{\downarrow0.1}$ |
| | CD | $77.5_{\downarrow7.6}$ | $80.9_{\downarrow4.2}$ | $81.5_{\downarrow3.6}$ | $82.7_{\downarrow2.4}$ | $83.4_{\downarrow1.7}$ | $84.9_{\downarrow0.2}$ | $14.7_{\downarrow11.6}$ | $18.0_{\downarrow8.3}$ | $20.8_{\downarrow5.5}$ | $21.9_{\downarrow4.4}$ | $25.1_{\downarrow1.2}$ | $26.0_{\downarrow0.3}$ |
| | Herding | $63.6_{\downarrow21.5}$ | $72.0_{\downarrow13.1}$ | $73.9_{\downarrow11.2}$ | $76.9_{\downarrow8.2}$ | $82.2_{\downarrow2.9}$ | $84.7_{\downarrow0.4}$ | $8.1_{\downarrow18.2}$ | $10.6_{\downarrow15.7}$ | $11.7_{\downarrow14.6}$ | $13.7_{\downarrow12.6}$ | $17.2_{\downarrow9.1}$ | $22.6_{\downarrow3.7}$ |
| | K-Means | $61.8_{\downarrow23.3}$ | $68.5_{\downarrow16.6}$ | $74.9_{\downarrow10.2}$ | $78.5_{\downarrow6.6}$ | $81.4_{\downarrow3.7}$ | $83.7_{\downarrow1.4}$ | $12.8_{\downarrow13.5}$ | $16.7_{\downarrow9.6}$ | $19.6_{\downarrow6.7}$ | $21.4_{\downarrow4.9}$ | $24.1_{\downarrow2.2}$ | $25.8_{\downarrow0.5}$ |
| | Least Confidence | $79.2_{\downarrow5.9}$ | $81.0_{\downarrow4.1}$ | $82.4_{\downarrow2.7}$ | $82.8_{\downarrow2.3}$ | $83.2_{\downarrow1.9}$ | $85.1_{\downarrow0.0}$ | $14.4_{\downarrow11.9}$ | $19.2_{\downarrow7.1}$ | $21.6_{\downarrow4.7}$ | $23.2_{\downarrow3.1}$ | $25.7_{\downarrow0.6}$ | $26.0_{\downarrow0.3}$ |
| | Entropy | $78.7_{\downarrow6.4}$ | $81.1_{\downarrow4.0}$ | $81.3_{\downarrow3.8}$ | $82.4_{\downarrow2.7}$ | $84.3_{\downarrow0.8}$ | $85.2_{\uparrow0.1}$ | $14.6_{\downarrow11.7}$ | $18.4_{\downarrow7.9}$ | $21.4_{\downarrow4.9}$ | $23.2_{\downarrow3.1}$ | $25.5_{\downarrow0.8}$ | $26.7_{\uparrow0.4}$ |
| | Forgetting | $80.0_{\downarrow5.1}$ | $81.6_{\downarrow3.5}$ | $82.9_{\downarrow2.2}$ | $83.8_{\downarrow1.3}$ | $84.7_{\downarrow0.4}$ | $84.9_{\downarrow0.3}$ | $15.3_{\downarrow11.0}$ | $18.9_{\downarrow7.4}$ | $21.3_{\downarrow5.0}$ | $22.3_{\downarrow4.0}$ | $25.3_{\downarrow1.0}$ | $26.1_{\downarrow0.2}$ |
| | GraNd-4 | $77.5_{\downarrow7.6}$ | $81.2_{\downarrow3.9}$ | $81.7_{\downarrow3.4}$ | $82.8_{\downarrow2.3}$ | $83.7_{\downarrow1.4}$ | $84.5_{\downarrow0.6}$ | $14.7_{\downarrow11.6}$ | $18.4_{\downarrow7.9}$ | $21.1_{\downarrow5.2}$ | $22.6_{\downarrow3.7}$ | $25.5_{\downarrow0.8}$ | $26.2_{\downarrow0.1}$ |
| | GraNd-20 | $80.1_{\downarrow5.0}$ | $82.5_{\downarrow2.6}$ | $83.0_{\downarrow2.1}$ | $83.9_{\downarrow1.2}$ | $84.7_{\downarrow0.4}$ | $84.9_{\downarrow0.2}$ | $15.8_{\downarrow10.5}$ | $19.4_{\downarrow6.9}$ | $22.0_{\downarrow4.3}$ | $23.1_{\downarrow3.2}$ | $25.7_{\downarrow0.6}$ | $26.0_{\downarrow0.3}$ |
| | DeepFool | $76.8_{\downarrow8.3}$ | $80.9_{\downarrow4.2}$ | $81.5_{\downarrow3.6}$ | $82.0_{\downarrow3.1}$ | $83.1_{\downarrow2.0}$ | $84.6_{\downarrow0.5}$ | $13.9_{\downarrow12.4}$ | $17.5_{\downarrow8.8}$ | $20.9_{\downarrow5.4}$ | $22.2_{\downarrow4.1}$ | $24.9_{\downarrow1.4}$ | $25.9_{\downarrow0.4}$ |
| | Craig | $76.5_{\downarrow8.6}$ | $80.8_{\downarrow4.3}$ | $81.3_{\downarrow3.8}$ | $82.5_{\downarrow2.6}$ | $83.8_{\downarrow1.3}$ | $85.0_{\downarrow0.1}$ | $14.5_{\downarrow11.8}$ | $18.7_{\downarrow7.6}$ | $21.3_{\downarrow5.0}$ | $22.9_{\downarrow3.4}$ | $25.1_{\downarrow1.2}$ | $26.0_{\downarrow0.3}$ |
| | Glister | $80.9_{\downarrow4.2}$ | $82.3_{\downarrow2.8}$ | $83.4_{\downarrow1.7}$ | $84.0_{\downarrow1.1}$ | $84.9_{\downarrow0.2}$ | $85.2_{\uparrow0.1}$ | $15.5_{\downarrow10.8}$ | $18.8_{\downarrow7.5}$ | $21.6_{\downarrow4.7}$ | $23.2_{\downarrow3.1}$ | $25.3_{\downarrow1.0}$ | $26.1_{\downarrow0.2}$ |
| | Influence | $76.5_{\downarrow8.6}$ | $80.5_{\downarrow4.6}$ | $81.7_{\downarrow3.4}$ | $82.5_{\downarrow2.6}$ | $83.4_{\downarrow1.7}$ | $84.2_{\downarrow0.9}$ | $13.7_{\downarrow12.6}$ | $17.9_{\downarrow8.4}$ | $20.5_{\downarrow5.8}$ | $22.1_{\downarrow4.2}$ | $24.5_{\downarrow1.6}$ | $25.4_{\downarrow0.9}$ |
| | EL2N-20 | $79.8_{\downarrow5.3}$ | $82.0_{\downarrow3.1}$ | $83.5_{\downarrow1.6}$ | $84.0_{\downarrow1.1}$ | $85.4_{\uparrow0.3}$ | $85.1_{\downarrow0.0}$ | $14.7_{\downarrow11.6}$ | $19.1_{\downarrow7.2}$ | $21.7_{\downarrow4.6}$ | $22.5_{\downarrow3.8}$ | $25.5_{\downarrow0.8}$ | $26.1_{\downarrow0.2}$ |
| | DP | $77.9_{\downarrow7.2}$ | $80.1_{\downarrow5.0}$ | $82.5_{\downarrow2.6}$ | $83.7_{\downarrow1.4}$ | $84.6_{\downarrow0.5}$ | $85.0_{\downarrow0.1}$ | $14.1_{\downarrow12.2}$ | $18.2_{\downarrow8.1}$ | $20.9_{\downarrow5.4}$ | $22.8_{\downarrow3.5}$ | $25.1_{\downarrow1.2}$ | $25.9_{\downarrow0.4}$ |
| Dynamic | Soft Random | $82.3_{\downarrow2.8}$ | $83.7_{\downarrow1.4}$ | $84.3_{\downarrow0.8}$ | $84.6_{\downarrow0.5}$ | $85.0_{\downarrow0.1}$ | $85.1_{\downarrow0.0}$ | $16.1_{\downarrow10.2}$ | $19.2_{\downarrow7.1}$ | $21.0_{\downarrow5.3}$ | $22.3_{\downarrow4.0}$ | $24.2_{\downarrow2.1}$ | $25.4_{\downarrow0.9}$ |
| | $\epsilon$-greedy | $82.5_{\downarrow2.6}$ | $83.2_{\downarrow1.9}$ | $83.7_{\downarrow1.4}$ | $84.1_{\downarrow1.0}$ | $84.8_{\downarrow0.3}$ | $85.1_{\downarrow0.0}$ | $16.5_{\downarrow9.8}$ | $19.8_{\downarrow6.5}$ | $20.3_{\downarrow6.0}$ | $21.5_{\downarrow4.8}$ | $23.8_{\downarrow2.5}$ | $25.2_{\downarrow1.1}$ |
| | UCB | $82.6_{\downarrow2.5}$ | $83.0_{\downarrow2.1}$ | $83.5_{\downarrow1.6}$ | $83.9_{\downarrow1.2}$ | $84.5_{\downarrow0.6}$ | $84.7_{\downarrow0.4}$ | $16.7_{\downarrow9.6}$ | $20.2_{\downarrow6.1}$ | $22.0_{\downarrow4.3}$ | $23.5_{\downarrow2.8}$ | $24.9_{\downarrow1.4}$ | $26.1_{\downarrow0.2}$ |
| | InfoBatch[1] | $82.9_{\downarrow2.2}$ | $83.5_{\downarrow1.6}$ | $84.4_{\downarrow0.7}$ | $84.9_{\downarrow0.2}$ | $85.4_{\uparrow0.3}$ | $85.2_{\uparrow0.1}$ | $19.9_{\downarrow6.4}$ | $22.8_{\downarrow3.5}$ | $24.5_{\downarrow1.8}$ | $25.5_{\downarrow0.8}$ | $26.8_{\uparrow0.5}$ | $27.0_{\uparrow0.7}$ |
| | MolPeg | $\mathbf{83.7}_{\downarrow1.4}$ | $\mathbf{84.8}_{\downarrow0.3}$ | $\mathbf{85.3}_{\uparrow0.2}$ | $\mathbf{85.5}_{\uparrow0.4}$ | $\mathbf{86.0}_{\uparrow0.9}$ | $\mathbf{85.6}_{\uparrow0.5}$ | $\mathbf{20.7}_{\downarrow5.6}$ | $\mathbf{23.9}_{\downarrow2.4}$ | $\mathbf{25.6}_{\downarrow0.7}$ | $\mathbf{26.4}_{\uparrow0.1}$ | $\mathbf{26.8}_{\uparrow0.5}$ | $\mathbf{27.0}_{\uparrow0.7}$ |
| | Whole Dataset | 85.1±0.3 | | | | | | 26.3±0.1 | | | | | |

[1] To make a fair comparison, we remove the annealing operation in the InfoBatch, since it uses the full dataset for training at later epochs.

**Training details.** We adhere to the settings proposed by [41] for our experiments. In classification tasks, the dataset is randomly split, with an 80%/10%/10% partition for training, validation and testing, respectively. In regression tasks, the QM9 dataset is divided into 110K molecules for training, 10K for validation, and another 10K for testing. The Adam optimizer [44] is employed for training with a batch size of 256. For classification tasks, the learning rate is set at 0.001 and we opt against using a scheduler. For regression tasks, we align with the original experimental settings of PaiNN and SchNet, setting the learning rate to $5 \times 10^{-4}$ and incorporating a cosine annealing scheduler.

**Evaluation protocols.** We conduct a series of experiments between model performance and varied data quantities. Specifically, we divide the pruning ratio into six proportional subsets: [20%, 40%, 60%, 70%, 80%, 90%], and for each configuration, we randomly select five seeds and report the mean performance. For HIV datasets, performance is measured using the Area Under the ROC-Curve (ROC-AUC), while reporting the performance on PCBA in terms of Average Precision (AP) —higher values in both metrics indicate better performance. When assessing quantum property predictions in the QM9 dataset, the Mean Absolute Error (MAE) is used as the performance metric, with lower values indicating better accuracy. The additional results on MUV dataset are demonstrated in Table 5.

## 5 Empirical Studies

### 5.1 Empirical analysis on classification tasks

Our empirical studies for classification tasks utilize the 2D graph modality as the input. We employ GIN as the backbone model and adopt GraphMAE [45] for model pre-training on the PCQM4Mv2 dataset. For a comprehensive comparison, we select the following two groups of DP methods as primary baselines in our experiments: static DP and dynamic DP, following [46]. The majority of previous methods fall into the former group, from which we select 14 competitive and classic DP methods as baselines, i.e., hard random pruning, CD [47], Herding [18], K-means [35], Least

Table 2: The performance comparison to state-of-the-art methods on QM9 dataset in terms of MAE (↓). We highlight the best- and the second-performing results in **boldface** and underlined, respectively.

| Dataset | QM9-U0 (meV) | | | | | | QM9-Zpve (meV) | | | | | |
|---|---|---|---|---|---|---|---|---|---|---|---|---|
| Pruning Ratio % | 90 | 80 | 70 | 60 | 40 | 20 | 90 | 80 | 70 | 60 | 40 | 20 |
| Random | **85.0** | **45.7** | 34.2 | 30.9 | 19.2 | 15.7 | **4.94** | **3.09** | 2.53 | 2.26 | 1.93 | 1.65 |
| DP | 136.0 | 68.5 | 39.8 | 32.3 | 20.8 | 16.1 | 8.56 | 6.29 | 3.62 | 2.36 | 2.05 | 1.68 |
| InfoBatch | 116.0 | 57.0 | 36.4 | 30.1 | 20.4 | 15.6 | 6.26 | 4.61 | 3.22 | 2.34 | 1.91 | 1.64 |
| MolPeg | 92.4 | 48.2 | **32.4** | **26.1** | **17.7** | **14.3** | 5.40 | 3.18 | **2.51** | **2.24** | **1.86** | **1.62** |

Confidence [48], Entropy [48], Forgetting [16], GraNd [19], EL2N [19], DeepFool [49], Craig [50], Glister [51], Influence [14] and DP [15]. Since dynamic pruning remains a niche topic, we identify four methods, to the best of our knowledge, to serve as baselines, i.e., soft random pruning, $\epsilon$-greedy [52], UCB [52] and InfoBatch [46], with MolPeg also falling into this category. Please refer to Appendix C for a more detailed introduction to the baselines.

**Performance comparison.** Empirical results for DP methods are presented in Table 1. Our systematic study suggests the following trends: *(i) Dynamic DP strategies significantly outperform static DP strategies.* Soft random, as a fundamental baseline in dynamic DP, consistently outperforms the baselines of static groups across almost all pruning ratios, even surpassing some strong competitors such as Glister and GraNd. We also observe that the performance advantage of dynamic DP becomes more pronounced when the pruning ratio is relatively large. Intuitively, compared to fixing a subset for training, dynamic pruning can perceive the full dataset during training, thereby possessing a larger receptive field and naturally yielding better performance. As more data samples are retained, the ability of both groups to perceive the full training set converges, leading to smaller performance differences between them. *(ii) MolPeg achieves the state-of-the-art performance across all proportions.* On the HIV dataset, we can achieve nearly lossless pruning by removing 80% of the samples, surpassing other baseline methods significantly. Similarly, on the larger-scale PCBA dataset, we can still achieve lossless pruning by removing 60% of the data. *(iii) MolPeg brings superior generalization performance compared to fine-tuning on the full dataset.* For example, on the HIV dataset, we achieve an ROC-AUC performance of 86 when pruning 40% of the data, surpassing the 85.1 achieved with training on the full dataset. This indicates that appropriate data pruning can better aid model generalization given a pre-trained model. However, as more downstream data is introduced, the improvement brought by our method diminishes, as shown by the 20% pruning proportion, due to introducing data samples that hinder model generalization. More empirical results on MUV dataset

**Efficiency comparison**. In addition to performance, time efficiency is another crucial indicator for DP. We conduct a performance-efficiency comparison of various DP methods on the HIV dataset at a 60% pruning ratio, as shown in Figure 4. We define *time efficiency* as the reciprocal of the runtime multiplied by 1000. A higher value of this metric indicates greater efficiency. We can observe that despite MolPeg experiencing slight efficiency loss compared to random pruning, it demonstrates superior pruning performance. Compared to the current SOTA baseline model, Info-Batch, our method achieves better model generalization with comparable efficiency. Conversely, static pruning methods incur 1.6x to 2.1x greater time costs than random pruning, with model performance stagnating or declining. This underscores that MolPeg achieves superior performance with minimal efficiency costs. Despite increased memory usage introduced by the reference model, EMA is commonly used to stabilize molecular training, which allows our method to utilize EMA-saved models without added memory overhead.

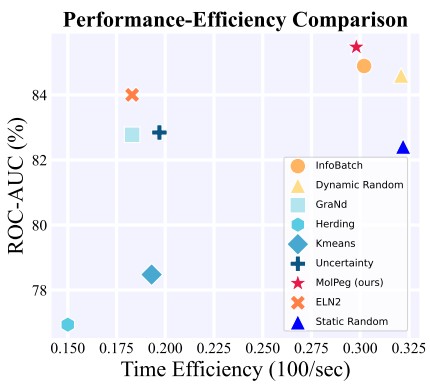

Figure 4: Performance and efficiency comparison between different DP methods. Pretrained models are fine-tuned on the HIV dataset at a 60% pruning ratio.

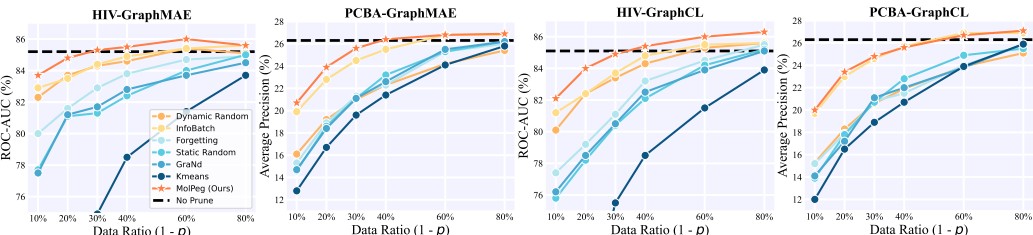

Figure 5: Data pruning trajectory given by downstream performance (%). Here the source models are pretrained on the PCQM4Mv2 dataset with GraphMAE and GraphCL strategies, respectively.

## 5.2 Results on QM9 dataset

Since regression is another common type of downstream molecular task, we also present the empirical results of `MolPeg` on two properties using the `QM9` dataset, alongside comparisons with state-of-the-art methods. To ensure a fair comparison of experimental results, we employ the commonly used 3D geometry modality for modeling. We adopt GeoSSL [53] as the pretraining strategy and PaiNN as the backbone model, following the settings outlined by Liu et al [53]. Empirical results are presented in Table 2. It can be observed that `MolPeg` consistently outperforms other DP methods. However, all DP methods unexpectedly demonstrate inferior performance than random pruning in certain pruning ratios (80% and 90%). We speculate this phenomenon is attributed to the PCQM4Mv2 dataset used for pre-training and the QM9 dataset having a close match in the distribution patterns of molecular features. Thus, any non-uniform sampling methods would lead to biased data pruning which exacerbates distribution shift and hinders domain generalization.

## 5.3 Sensitivity Analysis

We further conduct extensive sensitivity analysis to validate the robustness of `MolPeg` across different pre-training strategies, molecular modalities, pre-training datasets and hyperparameter choices. All experiments below are conducted on the HIV dataset.

**Robustness evaluation across pretraining strategies.** Given that `MolPeg` primarily targets scenarios involving pre-trained models, it is necessary to compare its robustness when applied with different pre-training strategies. Without loss of generality, we select two representative pre-training strategies: generative self-supervised learning (SSL) and contrastive self-supervised learning, both of which dominate the field of molecular pre-training. Specifically, in addition to the results based on Graph-MAE [45] (generative SSL) presented in Table 1, we also conduct experiments based on GraphCL (contrastive SSL) [54] whose results are shown in Figure 5. We can observe that `MolPeg` achieves optimal performance on both pre-training methods across different pruning ratios. Promisingly, it demonstrates better model generalization than training on the full dataset, indicating insensitivity to pre-training strategies of our proposed framework, thus allowing for convenient plug-in application to other pre-trained models in different molecular tasks.

**Robustness evaluation across modalities.** The selection of molecular modality has long been a contentious issue in the field. To validate the effectiveness of `MolPeg` across different molecular modalities, we present a comparison of pruning results using 3D geometry in the HIV dataset as shown in Table 3.

Table 3: Performance with 3D modality on HIV dataset.

| Pruning Ratio % | 60 | 40 | 20 |
|---|---|---|---|
| Random Pruning | $80.1_{\downarrow 1.3}$ | $80.8_{\downarrow 0.6}$ | $81.2_{\downarrow 0.2}$ |
| `MolPeg` | $81.9_{\uparrow 0.5}$ | $82.3_{\uparrow 0.9}$ | $82.2_{\uparrow 0.8}$ |
| Whole Dataset | $81.4 \pm 1.7$ | | |

We pretrain the SchNet [43] on the PCQM4Mv2 dataset, and keep other settings the same as in Section 4.2. It is evident from the results that the `MolPeg` framework, consistent with the conclusions drawn in Section 5.1, continues to outperform dynamic random pruning and enhance the model generalization ability. At a 40% pruning ratio, `MolPeg` also surpasses the performance achieved with training on the full dataset. This demonstrates the robustness of our proposed DP method across molecular modalities.

**Robustness evaluation across pretaining datasets.** In source-free transfer learning, pretrained model is a hard-encoded module, and their variations naturally lead to performance changes. Therefore,

it is necessary to evaluate the robustness of `MolPeg` when pretrained with different pre-training datasets. We conduct additional experiments on the HIV dataset using two pretrained models of varying quality, obtained from the ZINC15 [55] and QM9 datasets, respectively. Compared to the PCQM4Mv2 dataset used in the Section 5.1, these two datasets are smaller in scale and exhibit more pronounced distribution shifts, resulting in poorer pretraining quality. We observe the following trends from Table 5: (i) In the case of fine-tuning on the entire dataset, models pretrained on ZINC15 and QM9 show significantly inferior performance, even worse than training from scratch, indicating poor quality of pretrained models. (ii) `MolPeg` still achieves the best performance with these two pretrained models. This demonstrates the robustness of `MolPeg` across pretraining datasets.

**How to choose $\beta$.** Since EMA is a crucial component of our framework, it is necessary to evaluate how to choose a proper $\beta$. We conduct an empirical analysis on the HIV dataset across three pruning ratios, i.e., [0.1, 0.4, 0.8], and consider a candidate list covering the value ranges of $\beta$: [0.001, 0.01, 0.1, 0.5, 0.9]. Intuitively, a smaller $\beta$ implies a slower parameter update pace in the reference model. When $\beta = 0$, it signifies using a frozen pre-trained model as the reference. The experimental results corresponding to the variation of $\beta$ are illustrated in Figure 6. Empirical results indicate that the

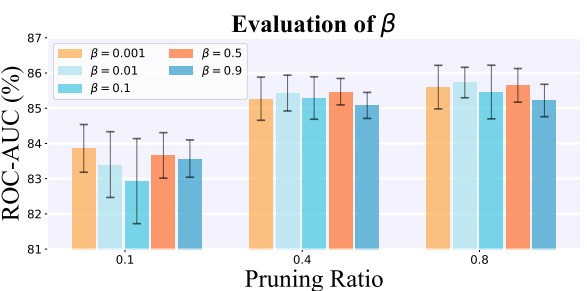

Figure 6: Performance bar chart of different choices of hyperparameter $\beta$ on HIV dataset. The error bar is measured in standard deviation and plotted in grey color.

overall performance shows only moderate sensitivity to parameter change. However, typically, when $\beta = 0.5$, the model tends to achieve better performance and smaller standard deviation. Hence, for our primary experiments, we opt to default to $\beta = 0.5$.

# 6 Conclusion

In this work, we propose `MolPeg`, a novel molecular data pruning framework designed to enhance generalization without the need for source domain data, thereby addressing the limitations of existing in-domain data pruning (DP) methods. Our approach leverages two models with different update paces to measure the informativeness of samples. Through extensive experiments across four downstream tasks involving both classification and regression tasks, we demonstrate that `MolPeg` not only achieves lossless pruning but also outperforms full dataset training in certain scenarios. This underscores the potential of `MolPeg` to optimize training efficiency and improve the generalization of pre-trained models in the molecular domain. Our contributions highlight the importance of considering source domain information in DP methods and pave the way for more efficient and scalable training paradigms in molecular machine learning.

**Broader impacts.** Given that our application tasks fall within the molecular domain, improper use of methods for tasks such as molecular property prediction may result in significant deviations. This could impact subsequent applications of the molecules in drug development or materials design, especially in predicting properties like toxicity and stability. We recommend further experimental validation of key molecules after using the model to ensure the reliability of the results. We provide further discussions in Appendix H.

# 7 Acknowledgements

This work is jointly supported by National Science and Technology Major Project (2023ZD0120901) and National Natural Science Foundation of China (62372454).

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

# A    Datasets and Tasks

In the following, we will elaborate on the adopted datasets and the statistics are summarized in Table 4.

Table 4: Statistics of datasets used in experiments.

|  | Dataset | Data Type | #Molecules | Avg. #atoms | Avg. #bonds | #Tasks | Avg. degree |
|---|---|---|---|---|---|---|---|
| Pre-training | PCQM4Mv2 | SMILES | 3,746,620 | 14.14 | 14.56 | - | 2.06 |
| Finetuning | HIV | SMILES | 41,127 | 25.51 | 27.47 | 1 | 2.15 |
|  | PCBA | SMILES | 437,929 | 25.96 | 28.09 | 92 | 2.16 |
|  | QM9-U0 | SMILES, 3D | 130,831 | 18.03 | 18.65 | 1 | 2.07 |
|  | QM9-ZPVE | SMILES, 3D | 130,831 | 18.03 | 18.65 | 1 | 2.07 |

- **PCQM4Mv2** is a quantum chemistry dataset curated by Hu et al. [56] based on the PubChemQC project [33]. It comprises 3,746,620 molecules and is extensively utilized in molecular pretraining tasks. We also adopt this widely recognized dataset for our molecular pretraining endeavors.

- **HIV** dataset is designed to evaluate the ability of molecular compounds to inhibit HIV replication [34] in a binary classification setting, consisting of 41,127 organic molecules.

- **PCBA** is a dataset consisting of biological activities of small molecules generated by high-throughput screening [37]. It contains 437,929 molecules with annotations of 92 classification tasks.

- **QM9** is a comprehensive dataset, structured for regression tasks, that provides geometric, energetic, electronic and thermodynamic properties for a subset of GDB-17 database, comprising 134 thousand stable organic molecules with up to nine heavy atoms [39]. In our experiments, we delete 3,054 uncharacterized molecules which failed the geometry consistency check [57]. We include the U0 and ZPVE in our experiment, which cover properties related to stability, and thermodynamics. These properties collectively capture important aspects of molecular behavior and can effectively represent various energetic and structural characteristics within the QM9 dataset.

- **MUV** (Maximum Unbiased Validation) group was selected from PubChem BioAssay via a refined nearest neighbor analysis approach, which is specifically designed for validation of virtual screening techniques [38].

# B    Computing infrastructures

**Software infrastructures.**    All of the experiments are implemented in Python 3.7, with the following supporting libraries: PyTorch 1.10.2 [58], PyG 2.0.3 [59], RDKit 2022.03.1 [60].

**Hardware infrastructures.**    We conduct all experiments on a computer server with 8 NVIDIA GeForce RTX 3090 GPUs (with 24GB memory each) and 256 AMD EPYC 7742 CPUs.

# C    Related work

Data pruning (DP) has been an ongoing research topic since the rise of deep learning. Traditional data pruning strategies often focus solely on the task dataset, exploring ways to represent the distribution of the entire dataset with fewer data points, thereby reducing training costs. However, with the recent advancements in transfer learning, focusing solely on the task dataset has become insufficient. Consequently, some data pruning strategies have been developed for transfer learning scenarios. We classify these strategies into in-domain data pruning and cross-domain data pruning.

**In-domain data pruning.** Most existing data pruning methods fall into this category. We further divide them into two groups: static data pruning and dynamic data pruning following [46]. Static data pruning aims to select a subset of data that remains unchanged throughout the training process, while dynamic data pruning methods consider that the optimal data subset evolves dynamically during training. Guo et al. [61] classify existing static data pruning methods based on their scoring function into the following categories: geometry [47, 62, 17, 63, 64], uncertainty [48], loss [16, 19, 46], decision boundary [49, 65], gradient matching [66, 50], bilevel optimization [13, 51, 67],

submodularity [68–70], and proxy [71, 48]. Despite dynamic data pruning is still in its early stages, it has demonstrated superior performance. Raju et al. [52] propose two dynamic pruning methods called UCB and $\epsilon$-greedy. These methods define an uncertainty value and calculate the estimated moving average. During each pruning period, $\epsilon$-greedy or UCB is used to select a fraction of the samples with the highest scores, and training is then conducted on these selected samples for that period. Recently, InfoBatch [46] achieves lossless pruning based on loss distribution and rescales the gradients of the remaining samples to approximate the original gradient. However, all of these methods place much emphasis on the target domain while ignoring the widespread use of transfer learning.

**Cross-domain data pruning.** We observe that with the use of pretraining, there is an additional source domain alongside the target domain. The key issue now is how to effectively utilize the information from both domains for data pruning in the context of transfer learning. To effectively address downstream tasks, a straightforward approach is to measure the distribution shift between the upstream and downstream data, and then prune the pretraining dataset to align its distribution with that of the downstream dataset [32, 31, 72, 22]. However, this method requires retraining the pretrained model for each different downstream task, which contradicts the intended *pretrain-finetune* paradigm. Therefore, we propose the problem of *source-free data pruning* which is aligned with practical usage of transfer learning.

# D    Backbone Model

## D.1    Embedding 2D graphs

Graph Isomorphism Network (GIN) [40] is a simple and effective model to learn discriminative graph representations, which is proved to have the same representational power as the Weisfeiler-Lehman test [73]. Recall that each molecule is represented as $\mathcal{G} = (\boldsymbol{A}, \boldsymbol{X}, \mathbf{E})$, where $\boldsymbol{A}$ is the adjacency matrix, $\boldsymbol{X}$ and $\mathbf{E}$ are features for atoms and bonds respectively. The layer-wise propagation rule of GIN can be written as:

$$\boldsymbol{h}_i^{(k+1)} = f_{\text{atom}}^{(k+1)}\left(\boldsymbol{h}_i^{(k)} + \sum_{j \in \mathcal{N}(i)}\left(\boldsymbol{h}_j^{(k)} + f_{\text{bond}}^{(k+1)}(E_{ij}))\right)\right), \tag{6}$$

where the input features $\boldsymbol{h}_i^{(0)} = \boldsymbol{x}_i$, $\mathcal{N}(i)$ is the neighborhood set of atom $v_i$, and $f_{\text{atom}}, f_{\text{bond}}$ are two MultiLayer Perceptron (MLP) layers for transforming atoms and bonds features, respectively. By stacking $K$ layers, we can incorporate $K$-hop neighborhood information into each center atom in the molecular graph. Then, we take the output of the last layer as the atom representations and further use the mean pooling to get the graph-level molecular representation:

$$\boldsymbol{z}^{\text{2D}} = \frac{1}{N}\sum_{i \in \mathcal{V}} \boldsymbol{h}_i^{(K)}. \tag{7}$$

## D.2    Embedding 3D geometries

**SchNet [43].**    We use the SchNet [43] as the encoder for the 3D geometries in HIV dataset. SchNet models message passing in the 3D space as continuous-filter convolutions, which is composed of a series of hidden layers, given as follows:

$$\boldsymbol{h}_i^{(k+1)} = f_{\text{MLP}}\left(\sum_{j=1}^{N} f_{\text{FG}}(\boldsymbol{h}_j^{(t)}, \boldsymbol{r}_i, \boldsymbol{r}_j)\right) + \boldsymbol{h}_i^{(t)}, \tag{8}$$

where the input $\boldsymbol{h}_i^{(0)} = \boldsymbol{a}_i$ is an embedding dependent on the type of atom $v_i$, $f_{\text{FG}}(\cdot)$ denotes the filter-generating network. To ensure rotational invariance of a predicted property, the message passing function is restricted to depend only on rotationally invariant inputs such as distances, which satisfying the energy properties of rotational equivariance by construction. Moreover, SchNet adopts radial basis functions to avoid highly correlated filters. The filter-generating network is defined as follow:

$$f_{\text{FG}}(\boldsymbol{x}_j, \boldsymbol{r}_i, \boldsymbol{r}_j) = \boldsymbol{x}_j \cdot e_k(\boldsymbol{r}_i - \boldsymbol{r}_j) = \boldsymbol{x}_j \cdot \exp(-\gamma\|\|\boldsymbol{r}_i - \boldsymbol{r}_j\|_2 - \mu\|_2^2). \tag{9}$$

Similarly, for non-quantum properties prediction concerned in this work, we take the average of the node representations as the 3D molecular embedding:

$$z^{\text{3D}} = \frac{1}{N} \sum_{i \in \mathcal{V}} h_i^{(K)}, \tag{10}$$

where $K$ is the number of hidden layers.

**PaiNN [42].** We use the PaiNN [42] as the encoder for the 3D geometries in QM9 dataset. PaiNN identify limitations of invariant representations in SchNet and extend the message passing formulation to rotationally equivariant representations, attaining a more expressive SE(3)-equivariant neural network model.

## E   Proof of Theoretical Analyses

**Assumption 1** (Slow parameter updating) *Assume the learning rate is small enough, so that the parameter update $\Delta\theta_t = \theta_{t+1} - \theta_t$ is small for every time step, i.e. $\|\Delta\theta_t\| \leq \epsilon$, $\forall t \in \mathbb{N}$, $\epsilon$ is a small constant.*

**Lemma 1.** *With the assumption of slow parameter update, we can prove that $\|\xi_t - \theta_t\| \leq \frac{1-\beta}{\beta}\epsilon$.*

*Proof.*

$$\begin{aligned} \xi_t - \theta_t &= (1-\beta)\xi_{t-1} - (1-\beta)\theta_t \\ &= (1-\beta)(\xi_{t-1} - \theta_{t-1}) - (1-\beta)\Delta\theta_{t-1} \\ &= -\sum_{j=1}^{t}(1-\beta)^j \Delta\theta_{t-j}. \end{aligned} \tag{11}$$

For the first two equations, we respectively use the definition of EMA parameter update in equation 1 and the definition of $\Delta\theta$. For the third equation, we iteratively employed the results from the previous two steps, along with the initial condition $\xi_0 = \theta_0$. With Assumption 1, we have

$$\|\xi_t - \theta_t\| \leq \sum_{j=1}^{t}(1-\beta)^j \epsilon \leq \frac{1-\beta}{\beta}\epsilon \tag{12}$$

$\square$

For the following results, we use the default setting in experiment $\beta = 0.5$, i.e. $\|\xi_t - \theta_t\| \leq \epsilon$.

**Proposition 1** (Interpretation of loss discrepancy) *With Assumption 1, the loss discrepancy can be approximately expressed by the dot product between the data gradient and the "EMA gradient":*

$$\mathcal{L}(x, \xi_t) - \mathcal{L}(x, \theta_t) = \alpha\nabla_\theta\mathcal{L}(x, \theta_t)v_t^{EMA} + O(\epsilon^2), \tag{13}$$

*where $v_t^{EMA}$ denotes $\sum_{j=1}^{t}(1-\beta)^j\nabla_\theta\mathcal{L}(\hat{\mathcal{D}}_{t-j}, \theta_{t-j})$, i.e. the weighted sum of the historical gradients, which we termed as "EMA gradient".*

*Proof.* From Lemma 1, since $\|\xi_t - \theta_t\|$ is small, we can use Taylor expansion of the loss function at $\theta_t$:

$$\begin{aligned} \mathcal{L}(x, \xi_t) - \mathcal{L}(x, \theta_t) &= \nabla_\theta\mathcal{L}(x, \theta_t)(\xi_t - \theta_t) + O(\|\xi_t - \theta_t\|^2) \\ &= \nabla_\theta\mathcal{L}(x, \theta_t)\sum_{j=1}^{t}(1-\beta)^j\nabla_\theta\mathcal{L}(\hat{\mathcal{D}}_{t-j}, \theta_{t-j}) + O(\|\epsilon\|^2), \end{aligned} \tag{14}$$

where we use equation 11 and the definition of online parameter update in equation 1. $\square$

**Proposition 2** (Gradient projection interpretation of `MolPeg`) *In the context of neglecting higher-order small quantities, we define $\mathcal{D}^+ \in \mathcal{D}$ and $\hat{\mathcal{D}}^+ \in \hat{\mathcal{D}}$ as samples that have positive dot products between the data gradient and the "EMA gradient", then*

$$\nabla_\theta\mathcal{L}(\hat{\mathcal{D}}^+, \theta) = \nabla_\theta\mathcal{L}(\mathcal{D}^+, \theta) + av^{EMA} + cv_\perp^{EMA}, a \geq 0, c \in \mathbb{R}. \tag{15}$$

*Similarly, we define $\mathcal{D}^- \in \mathcal{D}$ and $\hat{\mathcal{D}}^- \in \hat{\mathcal{D}}$ as samples that have negative dot products, then*

$$\nabla_\theta \mathcal{L}(\hat{\mathcal{D}}^-, \theta) = \nabla_\theta \mathcal{L}(\mathcal{D}^-, \theta) + b v^{EMA} + d v_\perp^{EMA}, b \leq 0, d \in \mathbb{R}. \tag{16}$$

*Equality holds if and only if the absolute loss discrepancy $|\mathcal{L}(x, \xi_t) - \mathcal{L}(x, \theta_t)|$ across $\mathcal{D}^+$ and $\mathcal{D}^-$ is uniform. This is a rare situation, and in such a case, our data selection strategy degenerates to random selection on $\mathcal{D}^+$ and $\mathcal{D}^-$.*

Discussion about $c$ and $d$ is provided after the proof.

*Proof.* In the context of neglecting higher-order small quantities, `MolPeg` selects data with large loss discrepancies, meaning large dot products by using Proposition 1. That is $\forall x \in \hat{\mathcal{D}}$ and $\forall x' \in \mathcal{D} \setminus \hat{\mathcal{D}}$, we have

$$|\nabla_\theta \mathcal{L}(x, \theta) \cdot v^{EMA}| \geq |\nabla_\theta \mathcal{L}(x', \theta) \cdot v^{EMA}|. \tag{17}$$

Then for samples in $\mathcal{D}^+$ and $\hat{\mathcal{D}}^+$, we have

$$\frac{1}{|\hat{\mathcal{D}}^+|} \sum_{x \in \hat{\mathcal{D}}^+} \nabla_\theta \mathcal{L}(x, \theta) \cdot v^{EMA} \geq \frac{1}{|\mathcal{D}^+|} \sum_{x \in \mathcal{D}^+} \nabla_\theta \mathcal{L}(x', \theta) \cdot v^{EMA} > 0 \tag{18}$$

That is $\nabla_\theta \mathcal{L}(\hat{\mathcal{D}}^+, \theta) \cdot v^{EMA} \geq \nabla_\theta \mathcal{L}(\mathcal{D}^+, \theta) \cdot v^{EMA} > 0$ for short. Thus when projecting $\nabla_\theta \mathcal{L}(\hat{\mathcal{D}}^+, \theta) - \nabla_\theta \mathcal{L}(\mathcal{D}^+, \theta)$ on $v^{EMA}$, the coefficient $a$ is $(\nabla_\theta \mathcal{L}(\hat{\mathcal{D}}^+, \theta) - \nabla_\theta \mathcal{L}(\mathcal{D}^+, \theta)) \cdot v^{EMA} \geq 0$.

Similarly,

$$\nabla_\theta \mathcal{L}(\hat{\mathcal{D}}^-, \theta) \cdot v^{EMA} \leq \nabla_\theta \mathcal{L}(\mathcal{D}^-, \theta) \cdot v^{EMA} < 0 \tag{19}$$

Then $c \triangleq (\nabla_\theta \mathcal{L}(\hat{\mathcal{D}}^-, \theta) - \nabla_\theta \mathcal{L}(\mathcal{D}^-, \theta)) \cdot v^{EMA} \leq 0$.

The condition for $a = 0$ ($c = 0$) is that the equality in equation 17 holds for samples in $\mathcal{D}^+$ ($\mathcal{D}^-$). □

Since our selection strategy does not constrain the direction perpendicular to the EMA gradient, we consider a simplified model where $b$ and $d$ are treated as random variables with an expectation of zero. Consequently, in the sense of expectation, equation 4 and equation 5 hold. The feasibility of this simplified model is demonstrated as follows. Assume that $\nabla_\theta \mathcal{L}(x, \theta) \cdot v_\perp^{EMA}$ for all samples are independent and identically distributed random variables with expectation $\mu$ and variance $\sigma^2$. When the sample sizes $|\mathcal{D}^+|$ and $|\hat{\mathcal{D}}^+|$ are sufficiently large, the central limit theorem implies that $\frac{1}{|\mathcal{D}^+|} \sum_{x \in \mathcal{D}^+} \nabla_\theta \mathcal{L}(x, \theta) \cdot v_\perp^{EMA}$ is approximately a Gaussian distribution $\mathcal{N}\left(\mu, \frac{\sigma^2}{|\mathcal{D}^+|}\right)$, and similarly, $\frac{1}{|\hat{\mathcal{D}}^+|} \sum_{x \in \hat{\mathcal{D}}^+} \nabla_\theta \mathcal{L}(x, \theta) \cdot v_\perp^{EMA}$ is approximately a Gaussian distribution $\mathcal{N}\left(\mu, \frac{\sigma^2}{|\hat{\mathcal{D}}^+|}\right)$. The expectation of their difference $\mathbb{E}c = \mathbb{E}\nabla_\theta \mathcal{L}(\hat{\mathcal{D}}^+, \theta) \cdot v_\perp^{EMA} - \mathbb{E}\nabla_\theta \mathcal{L}(\mathcal{D}^+, \theta) \cdot v_\perp^{EMA} = 0$. Similarly, we can prove $\mathbb{E}d = 0$.

## F  Connections to Existing DP Methods

### F.1  `MolPeg` & GraNd [19]

In the pretraining scenario, where the initialization is fixed, the GraNd score is defined as the norm of the gradient $\|\nabla_\theta \mathcal{L}(x, \theta_t)\|$. With Assumption 1, we can deduce $\|\xi_t - \theta_t\| \leq \epsilon$ as shown in equation 12, then the data selected by `MolPeg` satisfies $\delta \leq |\mathcal{L}(x, \theta_t) - \mathcal{L}(x, \xi_t)| = |\nabla_\theta \mathcal{L}(x, \theta_t)(\theta_t - \xi_t) + O(\epsilon^2)|$ $\leq \epsilon \|\nabla_\theta \mathcal{L}(x, \theta_t)\| + O(\epsilon^2)$. The data we select has a lower bound on the GraNd score $\|\nabla_\theta \mathcal{L}(x, \theta_t)\| \geq O(\frac{\delta}{\epsilon})$, making it more likely to be chosen by the GraNd score.

### F.2  `MolPeg` & Infobatch [46]

Our strategy employs relative loss scales rather than absolute values, enabling a more flexible adaptation for transfer scenarios. For simple downstream samples for pretraining model, where $\mathcal{L}(x, \xi_t)$ is small, both Infobatch and `MolPeg` eliminate samples with small online loss which are regarded as redundant for finetuning. However, for difficult samples for pretraining model, where $\mathcal{L}(x, \xi_t)$ is large, our method diverges from Infobatch by preserving the crucial samples for transfer learning.

### F.3  `MolPeg` & Forgetting [16]

If we consider classification tasks and use accuracy loss, our method tends to select samples near the classification boundary. This can be related to the forgetting method, which aims to select samples that have been forgotten (i.e., initially classified correctly and then incorrectly) multiple times. For simplicity, let's explain this in the context of binary classification under Assumption 1. Further assume the class prediction probability $f$ is $l$-Lipschitz continuous with respect to the parameters $\theta$, where $f = (f^{(0)}, f^{(1)})$ and $f^{(0)} + f^{(1)} = 1$, we have $\|f(x, \theta) - f(x, \xi)\| \leq l\epsilon$. The loss function $\mathcal{L}(x, \theta) = |\arg\max_i\{f(x, \theta)^{(i)}\} - y|, y \in \{0, 1\}$ is not continuous at the classification boundary where $f^{(0)}(x, \theta) = f^{(1)}(x, \theta) = 0.5$. Consequently, only when the sample is located near the classification boundary $f^{(0)}(x, \theta) \in (0.5 - l\epsilon/\sqrt{2}, 0.5 + l\epsilon/\sqrt{2})$, exhibit a non-zero loss discrepancy.

## G  Additional Empirical Results

### G.1  The frequency of sample usage

Figure 7: Statics of frequency of sample usage in HIV datasets. The x-axis represents the number of times samples are used throughout the entire training process

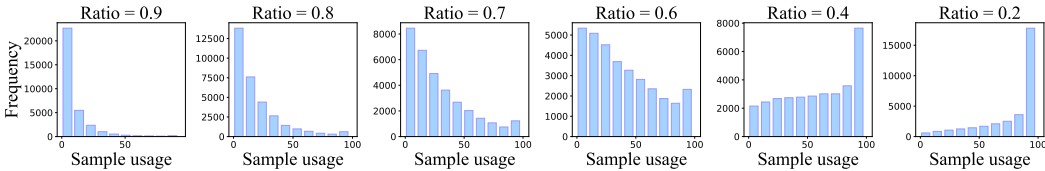

Unlike static DP, although we use a fixed proportion of samples in each iteration, we monitor the training dynamics of the entire dataset. Our method naturally avoids the issue of completely ignoring crucial samples, as almost **all samples** are used to varying degrees. Coordinating the use of samples in each iteration to achieve better generalization is the main contribution of `MolPeg`. In Figure 7, we visualize the frequency of sample usage in the HIV datasets, showing that almost all samples are used for training even with an aggressive pruning ratio adopted, with crucial samples being used more frequently.

### G.2  Robustness evaluation of different pretraining datasets

Table 5: The performance comparison on HIV with different pre-taining datasets of varying quality in terms of ROC-AUC (%, ↑)

| | Pre-train Dataset | QM9 (Low quality) | | | ZINC (medium quality) | | |
|---|---|---|---|---|---|---|---|
| | Pruning Ratio % | 90 | 70 | 40 | 90 | 70 | 40 |
| Static | Hard Random | $74.8_{\downarrow 7.9}$ | $78.1_{\downarrow 4.6}$ | $80.5_{\downarrow 2.2}$ | $78.2_{\downarrow 6.0}$ | $80.9_{\downarrow 3.3}$ | $83.1_{\downarrow 1.1}$ |
| | Forgetting | $76.5_{\downarrow 6.2}$ | $79.4_{\downarrow 3.3}$ | $80.9_{\downarrow 1.8}$ | $79.6_{\downarrow 4.6}$ | $81.5_{\downarrow 2.7}$ | $83.7_{\downarrow 0.5}$ |
| | GraNd | $77.3_{\downarrow 5.4}$ | $80.2_{\downarrow 2.5}$ | $80.8_{\downarrow 1.9}$ | $80.2_{\downarrow 4.0}$ | $81.7_{\downarrow 2.5}$ | $83.9_{\downarrow 0.3}$ |
| | Glister | $75.9_{\downarrow 6.8}$ | $78.6_{\downarrow 4.1}$ | $81.1_{\downarrow 1.6}$ | $78.6_{\downarrow 5.6}$ | $81.2_{\downarrow 3.0}$ | $83.5_{\downarrow 0.7}$ |
| Dynamic | Soft Random | $79.1_{\downarrow 3.6}$ | $81.3_{\downarrow 1.4}$ | $82.5_{\downarrow 0.2}$ | $81.8_{\downarrow 2.4}$ | $83.2_{\downarrow 1.0}$ | $84.0_{\downarrow 0.2}$ |
| | UCB | $79.5_{\downarrow 3.2}$ | $81.8_{\downarrow 0.9}$ | $82.5_{\downarrow 0.2}$ | $82.2_{\downarrow 2.0}$ | $83.1_{\downarrow 1.1}$ | $83.9_{\downarrow 0.3}$ |
| | InfoBatch | $80.5_{\downarrow 2.2}$ | $81.7_{\downarrow 1.0}$ | $83.1_{\uparrow 0.4}$ | $82.5_{\downarrow 1.7}$ | $83.8_{\downarrow 0.4}$ | $84.5_{\uparrow 0.3}$ |
| | `MolPeg` | $\mathbf{81.5}_{\downarrow 1.2}$ | $\mathbf{82.9}_{\uparrow 0.2}$ | $\mathbf{83.9}_{\uparrow 1.2}$ | $\mathbf{83.2}_{\downarrow 1.0}$ | $\mathbf{84.5}_{\uparrow 0.3}$ | $\mathbf{85.6}_{\uparrow 1.4}$ |
| | Whole Dataset | | 82.7±0.5 | | | 84.2±0.2 | |
| | Train from Scratch | | | 83.8±0.2 | | | |

### G.3  Results on MUV dataset

In Table 5, we have supplemented with additional experiments on the MUV dataset, following the same experimental setup described in Section 5.1. We observe that `MolPeg` still achieves state-of-the-art performance on the MUV dataset, further validating the effectiveness of our method.

Table 6: The performance comparison to state-of-the-art methods on MUV in terms of ROC-AUC (%, ↑).

| MUV | Static Pruning | | | | Dynamic Pruning | | | |
|---|---|---|---|---|---|---|---|---|
| Pruning Ratio % | Random | Forgetting | GraNd-20 | Glister | Random | UCB | InfoBatch | MolPeg |
| 90 | $73.8_{\downarrow 6.4}$ | $75.3_{\downarrow 4.9}$ | $75.6_{\downarrow 4.6}$ | $75.5_{\downarrow 4.7}$ | $76.3_{\downarrow 3.9}$ | $76.8_{\downarrow 3.4}$ | $78.0_{\downarrow 2.2}$ | $\mathbf{79.8}_{\downarrow 0.4}$ |
| 70 | $75.7_{\downarrow 4.5}$ | $76.8_{\downarrow 3.4}$ | $77.2_{\downarrow 3.0}$ | $77.6_{\downarrow 2.6}$ | $77.9_{\downarrow 2.3}$ | $78.2_{\downarrow 2.0}$ | $79.1_{\downarrow 1.1}$ | $\mathbf{80.7}_{\uparrow 0.5}$ |
| 40 | $78.4_{\downarrow 1.8}$ | $78.5_{\downarrow 1.7}$ | $77.9_{\downarrow 2.3}$ | $78.3_{\downarrow 1.9}$ | $79.0_{\downarrow 1.2}$ | $79.5_{\downarrow 0.7}$ | $80.5_{\uparrow 0.3}$ | $\mathbf{81.2}_{\uparrow 1.0}$ |
| Whole Dataset | 80.2±0.2 | | | | | | | |

## H   Discussions

**Limitations and future works.**   Our data pruning strategy is specifically designed for molecular downstream tasks, but source-free data pruning is a task setting with broad applications in other fields as well. For example, in large language models (LLMs) and heavy-weight models, pretraining data is often difficult for users to obtain or even kept confidential [74–77]. However, we have not validated our method in these more general scenarios. Therefore, verifying the effectiveness of MolPeg in more general tasks is one of our future research directions, e.g., graph, natural language and vision scenarios. Additionally, as the first work designed for the source-free data pruning setting, we have only made simple attempts at perceiving upstream and downstream knowledge via loss discrepancy. In the future, we will explore how to better utilize knowledge from both the source and target domains to achieve data pruning, which leaves significant potential to be explored.

