# OpenReview forum: "Beyond Efficiency: Molecular Data Pruning for Enhanced Generalization"
_NeurIPS.cc/2024/Conference — NeurIPS 2024 poster_

### Official Review · Reviewer_uU9Z · 2024-07-08

**Soundness:** 2
**Presentation:** 3
**Contribution:** 3
**Rating:** 6
**Confidence:** 3

**Summary:**

The paper presents a new framework MolPeg aimed at improving the efficiency and generalization of training models in molecular tasks using pretrained models. MolPeg introduces a novel DP technique that maintains two models with different update paces during training, leveraging the loss discrepancy between these models to score and select the most informative samples. This approach is applied in a source-free data pruning scenario, which does not require access to the original pretraining data. Extensive experiments across four molecular tasks demonstrate the effectiveness of MolPeg method.

**Strengths:**

1. The paper introduces a novel approach to data pruning by leveraging pretrained models and a loss discrepancy scoring mechanism.
2. The paper is well-organized, with a clear presentation of the problem, the proposed solution, and the experimental results.
3. The experiments are comprehensive, covering multiple datasets and tasks, and provide empirical evidence supporting the efficacy of MolPeg.

**Weaknesses:**

1. The effectiveness of MolPeg is highly dependent on the quality and suitability of the pretrained models utilized. If appropriate pretrained models are unavailable, the benefits of MolPeg may not be fully realized.
2. The proposed framework requires the maintenance and simultaneous updating of two models, which could increase both implementation complexity and computational overhead compared to simpler pruning methods.
3. Most DP methods demonstrate their effectiveness on widely-used datasets such as ImageNet. However, molecular datasets represent a more specialized field. It is crucial to test MolPeg with mainstream validation datasets to further demonstrate its effectiveness.

**Questions:**

Please refer to weakness 3, why not validate the effectiveness of MolPeg with mainstream datasets? Molecular datasets belong to a more specialized field and seem less persuasive compared to datasets like ImageNet.

**Limitations:**

Yes

---

> ### Author Rebuttal · Authors · 2024-08-06
>
> >**1. The effectiveness of MolPeg is highly dependent on the quality and suitability of the pretrained models utilized.**
>
> Thanks for your valuable comments. We agree with the reviewer’s point that the quality of pre-training can influence the effectiveness of our method. In source-free transfer learning, pretrained model is a hard-encoded module, and their variations naturally lead to performance changes. To address the reviewer's concern more thoroughly, we have conducted additional experiments on the HIV dataset using two pretrained models of different quality, obtained from the ZINC-100K and QM9 datasets, respectively. Compared to the PCQM4Mv2 dataset used in the main text, these two datasets are smaller in scale and exhibit more pronounced distribution shifts, resulting in poorer pretraining quality. The experimental results are shown in `Table 2` of the PDF file in the `General Response`.
>
> We observe the following trends: MolPeg still achieves the best performance with these two pretrained models, demonstrating that while pretraining quality is a key factor affecting performance, **MolPeg remains the most robust and effective method compared to existing DP strategies.** We again thank the reviewer for pointing out this issue and hope this explanation addresses your concerns.
>
>
>
> > **2. The proposed framework requires the maintenance and simultaneous updating of two models, which could increase both implementation complexity and computational overhead compared to simpler pruning methods.**
>
> Thanks for your valuable comments. The complexity is undoubtedly crucial for efficient learning, but we respectfully disagree with the reviewer regarding our method being more costly compared to existing pruning methods. It is worth noting that most existing DP methods are static and require scoring and ranking **on the full dataset before training**, which involves a high cost and computational overhead. We address the reviewer's concerns from two perspectives: complexity analysis and experimental validation.
>
> - **Complexity Analysis**: To compare the computational complexity of our method, MolPeg, with other data pruning methods, we use the following notations: $ N $ is the total number of data points, $ \delta $ is the pruning ratio, $ T $ is the total number of training epochs, $ t $ is the number of pre-scoring epochs required by other methods to determine scores, and $H_{fw}, H_{bw}$ denotes complexity of forward and backward operation. Note that compared to the TopK operation, the majority of the time consumption comes from the process of scoring the samples, as this process typically involves computing the loss, gradients, or even more complex metrics. Therefore, we have disregarded the complexity of the sorting process and focus on the forward and backward complexity.
>
>   |                     | MolPeg                              | Other methods                            |
>   | ------------------- | ----------------------------------- | ---------------------------------------- |
>   | **Forward passes**  | $ \mathcal{O}(2H_{fw}(N)) $         | $  \mathcal{O}(H_{fw}((t + T\delta)N)) $ |
>   | **Backward passes** | $  \mathcal{O}(H_{bw}(T\delta N)) $ | $  \mathcal{O}(H_{bw}((t + T\delta)N)) $ |
>
>   While MolPeg performs more forward passes due to processing two models as noticed by the reviewer, the computational bottleneck is the backward pass, which typically takes more than two times the effort of a forward pass [1]. Other methods require $ tN $ extra backward passes for pre-scoring, which outweighs their advantage in forward passes.
>
>   Therefore, despite the added forward pass complexity, **MolPeg is overall more efficient due to the significantly reduced number of backward passes**.
>
>   [1] PyTorch Distributed: Experiences on Accelerating Data Parallel Training. VLDB 2020.
>
> - **Experimental Validation**: In `section 5.1` of the manuscript, we have provided an experimental analysis of efficiency comparison. As seen in `Figure 4` of the manuscript, our method demonstrates significantly shorter runtime compared to previous DP methods and achieves better model performance. Compared to the latest dynamic pruning methods, although our runtime efficiency is slightly behind, we achieve better generalization performance, which is crucial for transfer learning scenarios. This minor efficiency compromise is acceptable.
>
> > **3. Molecular datasets represent a more specialized field. It is crucial to test MolPeg with mainstream validation datasets to further demonstrate its effectiveness.**
>
> Thanks for your constructive suggestion and helpful feedback. To further demonstrate MolPeg's effectiveness, we have provided the experimental results on CIFAR-10 and CIFAR-100 in `Table 3` of the rebuttal PDF file. For the experimental setup, we follow the settings of InfoBatch. However, since our method requires a pretrained model, we adopt MoCo strategy to pretrain the ResNet-18 on ImageNet, and then fine-tune it on the CIFAR dataset. As seen from the experimental results, **MolPeg still achieves the best results on most pruning ratios, further validating the effectiveness of our method across different domains.**
>
> Regarding the reviewer's concern about the used datasets, we would like to reclaim our research motivation. As mentioned in `General Response`, our research is problem-driven and targets on ubiquitous and unsolved problem in AI for Chemistry community. Moreover, molecular tasks represent a complex and comprehensive task system. Unlike traditional DP tasks in CV, which focus on image classification, we validate MolPeg on **both classification and regression tasks**, involving **various molecular modalities** and **diverse task types**, which makes our evaluation even more comprehensive than traditional DP approaches. We will add the above experimental results to the appendix of the revised manuscript and we hope this response addresses your concern.

---

> > ### Comment · Reviewer_uU9Z · 2024-08-09
> >
> > Hello,
> >
> > Thank you for your response, which I believe has addressed most of my questions. While I am not very familiar with "AI for Chemistry", I am leaning towards acceptance. I will finalize my ratings after further discussions with other reviewers and exchanging opinions with them.
> >
> > Best regards,
> >
> > Reviewer uU9Z

---

> > > ### Author Response · Authors · 2024-08-09
> > >
> > > Dear Reviewer uU9Z,
> > >
> > > We are delighted that most of your concerns have been addressed and truly appreciate your openness towards accepting our work.
> > >
> > > We understand that the final rating will be made after further discussions with other reviewers, and we hope that our manuscript continues to stand out during this process. If there are any remaining questions or additional clarifications needed, please feel free to reach out to us. We are more than happy to provide further information.
> > >
> > > Thank you again for your time and consideration.
> > >
> > > Best regards,
> > >
> > > Authors

---

> > > ### Author Response · Authors · 2024-08-14
> > >
> > > Dear Reviewer uU9Z,
> > >
> > > We hope this message finds you well.
> > >
> > > We sincerely appreciate the time and effort you've dedicated to reviewing our submission and your thoughtful consideration of our rebuttal. Your feedback has been invaluable in helping us refine our work. We understand that you may still be discussing the paper with other reviewers. As the Author-Reviewer discussion phase is coming to a close, we would greatly appreciate **any updates you may have on the current ratings**. We would also appreciate any further comments and questions.
> > >
> > > Thank you once again for your careful consideration!
> > >
> > > Best regards,
> > >
> > > Authors

---

> ### Author Response · Authors · 2024-08-08
> **Correction of Typing Error in Complexity Analysis**
>
> Dear Reviewer uU9Z,
>
> During our proofreading process, we discovered a typing error in the time complexity of the forward passes for the MolPeg method. It should be $ \mathcal{O}(2H_{fw}(T\delta N)) $, while the textual analysis in the initial rebuttal is correct. We sincerely apologize for this oversight and invite you to refer to the corrected time complexity table below:
>
> |                     | MolPeg complexity                           | Other methods complexity                                    |
> | ------------------- | ------------------------------------------- | ----------------------------------------------------------- |
> | **Forward passes**  | $ \mathcal{O}(2H_{fw}(T\delta N)) $         | $ \mathcal{O}(H_{fw}((t + T\delta)N)) $                     |
> |                     | forwarding both online and reference models | forwarding online model in pre-scoring and training epochs |
> | **Backward passes** | $  \mathcal{O}(H_{bw}(T\delta N)) $         | $  \mathcal{O}(H_{bw}((t + T\delta)N)) $                    |
> |                     | backwarding only the online model | backwarding parameters in both pre-scoring and training epochs |
>
> We appreciate your understanding and attention to this correction.
>
> Best regards,
>
> Authors

---

### Official Review · Reviewer_mAK4 · 2024-07-09

**Soundness:** 4
**Presentation:** 4
**Contribution:** 3
**Rating:** 7
**Confidence:** 4

**Summary:**

By utilizing the pre-trained models, this paper presents a plug-and-play framework (MolPeg) to prune target data without source dataset. By maintaining two models with different updating paces during training, this paper introduces a novel scoring function to measure the informativeness of samples based on the loss discrepancy. The experimental results on 3 datasets shows the enhanced efficiency and superior generalization in transfer learning.

**Strengths:**

- This paper is well-written. Each component of the design space is carefully explained and well-presented.
- This method is easy to follow can be adapted to other tasks.
- Extensive experiments are conducted to verify the effectiveness of this method.

**Weaknesses:**

(Minor) Missing recent works: 1) static data pruning [1,2], 2) dynamic data pruning [3]

1. Active learning is a strong baseline for data subset selection. NeurIPS workshop, 2022

2. CCS: Coverage-centric Coreset Selection for High Pruning Rates. ICLR, 2023
3. Prioritized Training on Points that are Learnable, Worth Learning, and Not Yet Learnt. ICML, 2022

**Questions:**

See weakness.

**Limitations:**

See weakness.

---

> ### Author Rebuttal · Authors · 2024-08-06
>
> > **(Minor) Missing recent works: 1) static data pruning [1,2], 2) dynamic data pruning [3]**
>
> Thanks for your recognition of our work and constructive suggestions. We apologize for omitting any relevant works. For the related works provided by the reviewer, we have added experimental results on the HIV and PCBA datasets for the first two works as additional baselines (AL and CCS). The experimental results are shown below:
>
> | HIV Pruning ratio (%) | 90       | 80       | 70       | 60       | 40       | 20       |
> | --------------------- | -------- | -------- | -------- | -------- | -------- | -------- |
> | AL                    | 80.7     | 81.1     | 82.9     | 84.0     | 84.8     | 85.1     |
> | CCS                   | 81.5     | 82.3     | 83.8     | 84.2     | 85.0     | 85.2     |
> | MolPeG                | **83.7** | **84.8** | **85.3** | **85.5** | **86.0** | **85.6** |
>
> | PCBA Pruning ratio (%) | 90       | 80       | 70       | 60       | 40       | 20       |
> | ---------------------- | -------- | -------- | -------- | -------- | -------- | -------- |
> | AL                     | 15.2     | 19.2     | 20.9     | 22.5     | 25.2     | 26.2     |
> | CCS                    | 15.5     | 19.9     | 21.5     | 23.5     | 25.9     | 26.3     |
> | MolPeG                 | **20.7** | **23.9** | **25.6** | **26.4** | **26.8** | **27.0** |
>
> It can be observed that MolPeG still achieves the best performance with significant improvements.  We will add the above experimental results and empirical analysis to the appendix of the revised manuscript to enrich our research.
>
> For the last related work, due to the time constraint during the rebuttal period, we are unable to reproduce their results. However, we will also include it in the discussion of related works in the revised version. Finally, we appreciate your helpful comments. If you have any other questions, please feel free to let us know.

---

> > ### Comment · Reviewer_mAK4 · 2024-08-11
> >
> > Thanks for the response. I've read through other reviewers' feedback and responses as well. I have no more questions and will keep the score as it is.
> >
> > Best regards,
> > Reviewer mAK4

---

> > > ### Author Response · Authors · 2024-08-11
> > >
> > > We truly appreciate your constructive feedback and continuous support, which we believe has further improved our work.
> > >
> > > Thank you again for your time and consideration!
> > >
> > > Best wishes,
> > >
> > > Authors

---

### Official Review · Reviewer_zkte · 2024-07-10

**Soundness:** 3
**Presentation:** 3
**Contribution:** 2
**Rating:** 5
**Confidence:** 4

**Summary:**

The paper introduces MolPeg, a molecular data pruning framework designed to enhance generalization when applying data pruning to pretrained models for molecular tasks. MolPeg uses two models with different updating rates to develop a new scoring function that assesses the informativeness of data based on loss discrepancies.

**Strengths:**

1. By maintaining dual models that focus on both source and target domains and introducing a novel scoring function that selects both easy and hard samples, MolPeg achieves efficient, lightweight data pruning without the need for retraining.
2. The paper provides the code, ensuring the reproducibility of the method. I will attempt to run the code in the coming weeks and may modify my review comments as necessary.

**Weaknesses:**

1. Although the paper claims to be pioneering in applying data pruning to pretrained models, the motivation may require further exploration. Specifically, how can it ensure that OOD samples crucial for each task are not pruned, which could potentially undermine the very purpose of the pretrained model?
2. The definition of what constitutes a 'hard case' is unclear. Are these cases task-specific? If so, there's a risk that pruning might eliminate hard cases essential for certain tasks, affecting the model's comprehensiveness and utility.
3. The approach might exacerbate the 'molecular cliff' problem, where slight changes in molecular structure lead to significant changes in activity, and such nuances could be lost with aggressive data pruning in pre-training.
4. The experimental section is limited to only three commonly used molecular datasets. Other equally important datasets, such as MUV, were not included in the comparisons.

**Questions:**

None

**Limitations:**

It is suggested that the discussion of limitations be moved to the main text.

---

> ### Author Rebuttal · Authors · 2024-08-06
>
> > **1. Although the paper claims to be pioneering in applying data pruning to pretrained models, the motivation may require further exploration. Specifically, how can it ensure that OOD samples crucial for each task are not pruned, which could potentially undermine the very purpose of the pretrained model?**
>
> Thanks for your valuable feedback and insightful concerns. Preserving crucial samples is indeed a challenge for static DP methods in OOD scenarios. However, we would like to clarify that we employ a **dynamic DP strategy**. We invite the reviewer to refer to the `General Response` describing our pruning setup and the pseudo-code in the `Appendix G` for a better understanding. Below, we explain in detail why OOD samples crucial for each task are not pruned:
>
> - **Broader Receptive Field of Dynamic Pruning Strategy.** Unlike static DP, although we use a fixed proportion of samples in each iteration, we monitor the training dynamics of the entire dataset. Our method naturally avoids the issue of completely ignoring crucial samples, as almost all samples are used to varying degrees. Coordinating the use of samples in each iteration to achieve better generalization is the main contribution of MolPeg. In `Figure 1` of additional PDF file, we visualize the frequency of sample usage in the HIV datasets, showing that almost all samples are used for training even with aggressive pruning ratio, with crucial samples being used more frequently.
> - **Crucial OOD samples are essentially our hard samples.** Note that it is a contentious issue to determine whether or not a sample is crucial. Even in supervised training with the full dataset, the importance of different samples vary across epochs. In our scenario, crucial OOD samples can be considered as the ones that the training struggles with, corresponding to those with gradients opposite to the EMA gradient, as theoretically analyzed in `Proposition 2`. Therefore, we actually regard crucial OOD samples as an important category (hard cases) to preserve, rather than filtering them out.
>
> > **2. The definition of what constitutes a 'hard case' is unclear. Are these cases task-specific? If so, there's a risk that pruning might eliminate hard cases essential for certain tasks, affecting the model's comprehensiveness and utility.**
>
>
> We apologize for any lack of clarity in our descriptions. Below, we provide clearer explanations for hard and easy cases to address reviewer's concern, and we will also include these in the revised main text for better understanding.
>
> - **Hard cases** refer to samples that the model struggles with during optimization. These can also be understood as samples near the decision boundary of downstream tasks. As analyzed in lines 156-160 of our manuscript, these samples satisfy $\mathcal{L}(x,\theta_t)-\mathcal{L}(x,\xi_t)>0$，indicating that this epoch's optimization gave negative feedback compared to historical optimization.
> - Conversely, **easy cases** are samples that can be optimized very smoothly, leading to a continuous loss reduction, satisfying $\mathcal{L}(x,\theta_t)-\mathcal{L}(x,\xi_t)<0$.
>
> Moreover, we want to emphasize that neither hard nor easy cases are pre-defined; they are identified in real-time based on the loss discrepancy reflecting training dynamics, making them definitely task-specific since the loss value is closely related to the specific task. Regarding the reviewer's concern about the risk that eliminating hard cases, we believe this is similar to concern in `Weakness 1`. In our previous response, hard samples are actually crucial OOD samples. Therefore, **our method does not eliminate these cases but rather preserves them as informative ones.**
>
>
>
> > **3. The approach might exacerbate the 'molecular cliff' problem, where slight changes in molecular structure lead to significant changes in activity, and such nuances could be lost with aggressive data pruning in pre-training.**
>
>
> Thank you for your constructive feedback. We believe there might be some misunderstanding regarding our data pruning setup. As elaborated in the `General Response`, our data pruning is conducted on the downstream dataset, not during the pretraining stage.
>
> Furthermore, we would like to address the reviewer's concerns from an experimental perspective. If our method exacerbated the molecular cliff phenomenon in the pruned training set, the test performance in a random split setup would be poor. This is because retaining only these special cases during training would lead to a significant distribution shift between the training and test sets. However, **the experimental results in `Section 5` demonstrate SOTA data pruning performance**, and these **experiments were all conducted under the random split as mentioned in line 232 of our manuscript**. Therefore, we believe that the issue the reviewer is concerned about does not arise with our method.
>
>
>
> > **4. The experimental section is limited to only three commonly used molecular datasets. Other equally important datasets, such as MUV, were not included in the comparisons.**
>
> Thanks for your valuable comments to further enrich our empirical analysis. We have supplemented our work with additional experiments on the MUV dataset, following the same experimental setup described in the manuscript. Please refer to `Table 1` in the PDF file in the `General Response` for the results. We observe that MolPeg still achieves state-of-the-art performance on the MUV dataset, further validating the effectiveness of our method. In the revised version, we will include these additional experimental results in the appendix to enrich the experimental validation.
>
>
>
> > **5. It is suggested that the discussion of limitations be moved to the main text.**
>
> Thank you for pointing out this problem. In the revised version, we will follow the reviewer's suggestion and move the limitation to the main text.

---

> > ### Author Response · Authors · 2024-08-12
> >
> > Dear Reviewer zkte,
> >
> > We wanted to gently remind you that the deadline for the discussion phase is approaching, and we would greatly appreciate it if you could take a moment to review our responses.
> >
> > Your feedback is very valuable to us, and we are eager to hear your thoughts. If there are any additional concerns or points of clarification, we are more than happy to address them.
> >
> > Thank you for your time and consideration.
> >
> > Best wishes,
> >
> > Authors

---

> ### Comment · Area_Chair_RaHc · 2024-08-13
>
> Hi Reviewer zkte,
>
> Does the author’s response address your concerns? Please acknowledge that you have read the responses at your earliest convenience.
>
> Best wishes,
>
> AC

---

### Author Rebuttal · Authors · 2024-08-06

### **General Response**

We would like to thank all reviewers very much for their extensive reviews and constructive critiques. We are encouraged that reviewers find that our approach is efficient and lightweight (Reviewer zkte), that the experiments are comprehensive and verify the effectiveness (Reviewer uU9Z and mAK4), that the paper is well-organized with a clear presentation (Reviewers mAK4 and uU9Z).

However, we notice that the research motivation and data pruning setup have not been fully captured, leading to potential misunderstandings of some reviewers. Therefore, we would like to **restate our motivation and pruning setup** to address related concerns:

- **Motivation**: We point out the current need for efficient training in molecular modeling and attempt to improve training efficiency from the perspective of data pruning. Our exploration in the `Introduction` shows that **traditional DP methods fail** in the molecular domain due to significant distribution shifts, damaging the model generalization. Moreover, such distribution shift is inevitably due to the continual influx of novel molecular structures and functionalities in downstream tasks. In response to this phenomenon, we propose **the first source-free DP setup tailored for molecular domain**, which targets the main practical challenge in the field and represents the core motivation of our research.
- **Pruning Setup**: We want to emphasize that our approach involves a **dynamic data pruning on downstream datasets**, rather than static pruning on pretraining dataset. Unlike static pruning, which selects a fixed subset before training, our method dynamically selects the samples in each iteration. This means our approach **adapts in real-time based on training dynamics**, avoiding selection biases inherent in static DP methods and better catering to the specific task requirements.

### **Contents of PDF File**
Moreover, we have provided an extra **PDF file containing results and figures supporting our rebuttal arguments**. It should be noted that due to the time and page constraints during the rebuttal period, we were unable to supplement comparative experiments for all pruning ratios and all baselines. However, we selected 3 challenging aggressive pruning ratios and 8 competitive DP baselines for comparison. We believe these results could sufficiently validate the effectiveness of our method in the additional experiments. Below, we provide **a brief summary** of the content in the PDF for the reviewers' reference:

- **Figure 1** : We present the usage frequency statistics of samples when MolPeg is applied on the HIV dataset. The x-axis represents the number of times samples are used throughout the entire training process, and the y-axis represents the corresponding sample amounts. (for Reviewer `zkte` Weakness 1)
- **Table 1** : We have supplemented the pruning effectiveness of our method on the MUV dataset. (for Reviewer `zkte` Weakness 4)
- **Table 2** : We have supplemented the robustness of MolPeg's performance when using lower-quality pretrained models. These lower-quality pretrained models were obtained from the ZINC and QM9 datasets, which have more limited diversity and a smaller scale compared to the PCQM4Mv2 dataset used in the main text. (for Reviewer `uU9Z` Weakness 1)
- **Table 3** : We have supplemented MolPeg's performance on the classic image classification datasets—— CIFAR-10 and CIFAR-100. (for Reviewer `uU9Z` Weakness 3)

Finally, we appreciate all your helpful comments that strengthen the quality and clarity of our work. we hope the following responses address your concerns and we look forward to engaging in an active and productive discussion with the reviewers. If you have any other questions, please feel free to let us know.

---

### Comment · Area_Chair_RaHc · 2024-08-11

Dear Reviewers,

The deadline of reviewer-authors discussion is approaching. If you have not done so already, please check the rebuttal and provide your response at your earliest convenience.

Best wishes,

AC

---

### Decision · Program_Chairs · 2024-09-25

**Decision:**

Accept (poster)

**Comment:**

After discussion, all reviewers provide positive recommendations. In particular, all reviewer recognized that the proposed method is novel with comprehensive experiments. They also suggested several aspects where the paper can benefit from, including analysis of various pretrained models and experiments on mainstream validation sets. The authors have addressed most of these concerns properly during rebuttal, and  should include these results in the final version. Therefore, I recommend acceptance for this work.